# Effect of Gut Microbiota Alteration on Colorectal Cancer Progression in an In Vivo Model: Histopathological and Immunological Evaluation

**DOI:** 10.3390/cimb48010015

**Published:** 2025-12-23

**Authors:** Juliana Montoya Montoya, Elizabeth Correa Gómez, Jorge Humberto Tabares Guevara, Julián Camilo Arango Rincón, Tonny Williams Naranjo Preciado

**Affiliations:** 1Medical and Experimental Mycology Group, Corporación para Investigaciones Biológicas (CIB-UPB-UdeA-UDES), Carrera 72 A # 78B-141, Medellin 050034, Colombia; julianamontoya9923@gmail.com (J.M.M.); ecorrea84@gmail.com (E.C.G.); julian.arango@udea.edu.co (J.C.A.R.); 2Infettare Research Group, Universidad Cooperativa de Colombia, Av. Colombia #41-26, Medellin 050013, Colombia; jorge.tabares@udea.edu.co; 3Immunovirology Group, Universidad de Antioquia, Cl. 67 #53-108, Medellín 050010, Colombia; 4School of Microbiology, Universidad de Antioquia, Cl. 67 #53-108, Medellín 050010, Colombia; 5School of Health Sciences, Universidad Pontificia Bolivariana, Cq. 1 #70-01, Medellin 050034, Colombia

**Keywords:** colorectal cancer, gut microbiota, fecal microbiota transplantation, Balb/c mice

## Abstract

**Background/Objectives**: Colorectal cancer (CRC) is one of the leading causes of cancer-related mortality worldwide, with its development influenced by diet, obesity, and gut microbiota (GM) alterations. This study aimed to evaluate the impact of human fecal microbiota transplantation (FMT) on the progression of CRC in a murine model. **Methods**: CRC was chemically induced in BALB/c mice using azoxymethane/dextran sulfate sodium (AOM/DSS). Mice were transferred with GM via FMT and divided into two experimental groups according to the microbiota source (healthy donors or CRC patients). A positive control group (AOM/DSS without FMT) and a negative control group (no CRC induction or FMT) were included. Clinical parameters, histopathological analyses, and cytokine profiling were performed. **Results**: Mice receiving FMT, particularly from CRC patients, exhibited increased mitotic activity, dysplasia, neoplastic proliferation, structural alterations in the colon, and more pronounced GALT hyperplasia. At the immunological level, both FMT groups (healthy and CRC-derived) showed modulation of IL-1β, IL-4, IL-6, IL-10, IL-17A, and TNF-α compared to the positive control. **Conclusions**: Human GM transplantation modulated the colonic microenvironment through histopathological and immunological changes, influencing CRC progression in this murine model. These findings highlight the role of GM in shaping CRC development and suggest that human-derived microbiota may significantly impact tumor dynamics.

## 1. Introduction

Cancer is characterized by the uncontrolled proliferation of abnormal cells, which can invade adjacent tissues and form metastases in distant organs, particularly in advanced stages of the disease [1] Among the most common types, colorectal cancer (CRC) stands out for its global incidence and mortality, ranking third in incidence and second in mortality among both sexes in 2022 [2]. CRC originates in the colon—a section of the large intestine—or in the rectum, the canal that connects the colon to the anus [3,4].

The pathogenesis of CRC is a complex process resulting from the progressive accumulation of genetic and epigenetic alterations in the colonic epithelial tissue, leading to its transformation into adenocarcinoma [5]. This process can be divided into three main stages: initiation, promotion, and progression. Initiation is characterized by mutations induced by carcinogenic agents in the cells, followed by a prolonged promotion phase marked by abnormal cell proliferation, culminating in tumor progression and metastases [6]. These alterations result from the interaction of multiple factors, including genetic predisposition, environmental, dietary, and lifestyle-related factors [7].

Among the causes are modifiable factors such as physical inactivity, obesity, and inadequate dietary patterns, particularly those characterized by low intake of fiber, fruits, and vegetables [8], in addition to habits such as tobacco and alcohol consumption, among others [9]; and non-modifiable factors, including certain hereditary diseases such as familial adenomatous polyposis and Lynch syndrome, which together account for approximately 5–10% of CRC cases [4]. In this context, the gut microbiota (GM) represents a diverse community of microorganisms residing in the digestive tract, playing a crucial role in intestinal homeostasis and the development of CRC [10]. The gut microbiota is involved in the metabolism of indigestible food components, maintains intestinal barrier integrity, and modulates the immune response [11]. Imbalances in the gut microbiota can lead to chronic inflammation and the production of toxic metabolites [12], factors that contribute to colorectal carcinogenesis. Studies in murine models have demonstrated that alterations in the gut microbiota increase the incidence of colon tumors [13]. Moreover, the gut microbiota plays a crucial role in immune regulation by facilitating the differentiation of T and B cells and promoting the production of cytokines such as IL-17 and IL-23. It also modulates the balance between pro-inflammatory T helper 17 (Th17) cells and regulatory T (Treg) cells, which is essential for maintaining intestinal homeostasis and preventing excessive inflammation [13,14]. The composition of the gut microbiota in patients with colorectal cancer (CRC) differs markedly from that observed in healthy individuals, suggesting that microbial imbalance may be a key factor in CRC progression [15].

In this context, strategies aimed at modulating GM have gained relevance as potential therapeutic approaches [16]. One such strategy is fecal microbiota transplantation (FMT), which involves the administration of microbiota from a healthy donor to a recipient with intestinal dysbiosis. In preclinical models and emerging clinical studies, FMT has shown potential to restore microbial composition, improve immune responses, and modulate inflammatory processes associated with CRC [17]. This procedure has proven effective in various metabolic, infectious, and inflammatory disorders, such as including *Clostridioides difficile* infection (CDI) and inflammatory bowel disease (IBD). In cases of recurrent CDI, FMT interrupts the cycle of recurrence and achieves a resolution rate of up to 85% of patients [18] Similarly, the transferred microorganisms have shown beneficial effects in reducing inflammation in patients with IBD. A meta-analysis conducted in 122 patients reported an IBD remission rate of 45% after FMT, significantly higher than the 20% observed in the placebo group [19].

To study the influence of GM on CRC development, various animal models have been developed, including genetically modified models, xenotransplants, and chemically induced models [20]. Among the chemically induced models is the AOM/DSS model, a powerful initiation–promotion model based on the initial DNA damage induced by AOM. This compound acts as an alkylating agent, generating free radicals that bind to DNA and cause mutations [21,22]. Subsequently, repeated cycles of DSS are administered to induce colitis, allowing for the replication of the different stages of human CRC development within a relatively to be reproduced in a short period of time frame [21].

Although the GM of mice and humans differs significantly, human-to-mouse FMT could represent a valuable yet underexplored tool. This approach enables the replacement of the animal’s native microbiota through colonization with human-derivate microorganisms, preserving a diversity and profile like that of the donor. In this manner, a more physiological relevant model is established to investigate the role of human microbiota in diseases such as CRC [23,24]. Accordingly, this research aims to evaluate the histopathological and immunological effects of GM alteration on CRC progression using an in vivo model transplanted with human fecal microbiota.

## 2. Materials and Methods

### 2.1. Ethical Considerations

This study was conducted with the approval of the Institutional Bioethics Committee of the Universidad de Santander (Bucaramanga, Colombia), Record No. 16 dated 28 August 2023. All experiments were conducted in accordance with the ethical principles established by Law 84, enacted by the Congress of Colombia on 27 December 1989, and Resolution 8430 of 1993 from the Ministry of Health of Colombia. Likewise, the International Ethical Guidelines for Biomedical Research Involving Animals, developed by the Council for International Organizations of Medical Sciences (CIOMS) and established by UNESCO and the World Health Organization in 1949, were considered. The universal ethical principles and the internationally recognized “Three Rs” (Replacement, Reduction, and Refinement) principles were also followed.

### 2.2. Animals

Male and female BALB/c mice, 8 to 9 weeks old, weighing between 17 and 25 g (*n* = 22), were used. The animals were obtained from the animal facility of the Corporación para Investigaciones Biológicas (Medellin, Colombia) and maintained under controlled conditions: temperature of approximately 22 °C, 12 h light/dark cycles, and relative humidity between 60% and 70%. Additionally, the mice had ad libitum access to sterile water acidified with 0.01% HCl and standard rodent chow (LabDiet^®^, St. Louis, MO, USA).

### 2.3. Induction of Colorectal Cancer

The CRC model was chemically induced using AOM/DSS [20,25]. Briefly, animals received an intraperitoneal injection of AOM (Sigma-Aldrich, St. Louis, MO, USA) at a dose of 10 mg/kg in week 1. Subsequently, two cycles of DSS (MP Biomedicals, Santa Ana, CA, USA) at 2% in drinking water were administered for seven consecutive days during weeks 2 and 5 post-AOM injections. Throughout the experiment, the animals’ body weight was monitored. Two euthanasia time points were established for three animals per group, at the end of weeks 5 and 12.

### 2.4. Human Fecal Microbiota

To perform microbiota transfer or fecal microbiota transplantation (FMT) in the CCR mouse model, a spontaneously collected fecal sample from a designated healthy donor was utilized. The donor, a 62-year-old woman, was invited to participate in this study and provided informed consent, thereby voluntarily agreeing to take part (Appendix A: informed consent). A structured questionnaire was administered to assess the donor’s dietary habits and lifestyle (Appendix A: Fecal Donor Questionnaire). At the time of sample collection, the donor adhered to a balanced diet, regularly consuming fruits and vegetables, with a low intake of red and processed meats. She did not consume alcohol or tobacco and engaged in low to moderate physical activity. Additionally, the donor had no significant medical conditions and was not on any regular medications.

Additionally, a fecal sample from a donor diagnosed with colorectal cancer (CRC) was used for fecal microbiota transplantation (FMT). The donor was a 76-year-old male patient with stage III colorectal adenocarcinoma. His medical history included hypertension and type 2 diabetes mellitus, with no family history of cancer. Extensional studies, including abdominal and pelvic computed tomography, revealed cholelithiasis without signs of cholecystitis, a cystic lesion in the upper third of the right kidney, an enlarged prostate with early signs of bladder outlet obstruction, and diverticular disease of the sigmoid colon without complications. Furthermore, a small amount of free fluid was observed in the pelvis. Genetic testing identified RAS gene mutations but no evidence of mismatch repair deficiency.

### 2.5. Experimental Design

The 22 animals were randomly assigned to the following experimental groups (Figure 1):

Negative control: Mice without human FMT or CRC induction (*n* = 4).

Positive control: Mice without human FMT and CRC induction using AOM/DSS (*n* = 6).

FMT-CRC group: Mice receiving human FMT from a patient diagnosed with CRC and CRC induction using AOM/DSS (*n* = 6).

FMT-Healthy group: Mice receiving human FMT from a healthy donor and CRC induction using AOM/DSS (*n* = 6).

For clarity, all mice treated with AOM/DSS are collectively referred to as CRC-induced groups, which include the positive control, the FMT-CRC group, and the FMT-Healthy group. Mice not exposed to AOM/DSS are referred to as the negative control group.

### 2.6. Optimization and Validation of the Intestinal Cleansing Protocol and Human FMT in BALB/c Mice

To perform FMT to animals, an exhaustive literature review was conducted on the methodologies used for intestinal microbiota (IM) cleansing or depletion, as well as on the concentrations and frequencies employed for fecal microbiota transplantation (FMT) in mice (Table 1). This review aimed to verify that the methodology proposed for this study was appropriate and remained up to date. The analysis revealed a lack of consensus regarding intestinal cleansing and FMT procedures. Studies report the use of PEG and/or different antibiotics at various concentrations and for variable durations. Likewise, the concentration of the fecal material and the frequency of FMT differ among publications.

In this context and considering the advantages of the proposed methodology—based on the use of PEG to induce intestinal lavage—a pilot experiment was performed in which two mice were fasted for one hour. Subsequently, they received four doses of 200 µL of PEG at 42.5% (*w*/*v*), with 20 min intervals between each administration. The animals were then kept fasting for an additional four hours and were subsequently euthanized. Examination of the intestine confirmed complete evacuation, demonstrating the effectiveness of the procedure.

Regarding FMT, the literature indicates that most studies perform periodic booster administrations following the initial transfer. Therefore, in our model, FMT was administered every four weeks, maintaining intestinal lavage only for the first session to promote the colonization of human IM in the mouse.

### 2.7. Human Fecal Microbiota Transplantation in Mice

The method proposed by Wrzosek et al. was used with some modifications [24]. Briefly, prior to fecal microbiota transplantation (FMT), native intestinal microbiota was removed in both the FMT-Healthy and FMT-CRC groups. To achieve this, animals were fasted for one hour, followed by oral administration of four doses of polyethylene glycol (PEG4000) at 425 g/L (200 μL per dose) via orogastric gavage, with 20 min intervals between doses. Four hours after the last PEG4000 dose, animals were inoculated with a previously preserved fecal aliquot obtained from their respective donors, administering 200 μL of the 1% fecal suspension in phosphate-buffered saline (PBS, Sigma Aldrich, St. Louis, MO, USA) by oral gavage (week −1), after which food was reintroduced. This procedure was repeated every four weeks without intestinal lavage (weeks 4, 8, and 12).

### 2.8. Animal Euthanasia

Animal euthanasia was performed by CO_2_ inhalation at weeks 5 and 12 (W5 and W12, respectively). Subsequently, the large intestine was dissected, opened longitudinally, washed with PBS, measured, and weighed. The last 6 cm from the distal to the proximal end were collected. One portion was fixed in 10% buffered formalin (Sigma-Aldrich, St. Louis, MO, USA) for histopathological analysis, while the remaining portion was homogenized for cytokine measurement [25].

Due to the small size and fragility of colonic tissue in the negative control group, samples from individual animals were allocated exclusively to either histopathological analysis or cytokine quantification. Thus, at each time point (weeks 5 and 12), one animal was used for histopathological evaluation, and a different animal was used for cytokine analysis in the negative control group.

### 2.9. Histopathological Analysis

To evaluate the adenocarcinomatous process, a portion of the intestine fixed in 10% buffered formalin was used. The sample was sent to the Animal Pathology Laboratory of the Faculty of Agricultural Sciences at the University of Antioquia (Colombia), where it was embedded in paraffin and subsequently stained with Hematoxylin and Eosin. Histopathological analysis was conducted by expert veterinary pathologists, who assessed various parameters, including the degree of atrophy, hyperplasia, dysplasia, neoplastic proliferation, neoplastic infiltration, erosion, necrosis, apoptosis, and the level of tissue inflammation. These parameters were quantitatively scored on a scale from 0 to 3 (0: No lesion or absence of the condition, 1: Mild to moderate lesion, 2: Moderate lesion, and 3: Moderate to severe lesion).

### 2.10. Inflammatory Cytokines

A portion of the large intestine was immersed in 2 mL of sterile 1X PBS supplemented with 1% protease inhibitor cocktail solution (Sigma-Aldrich, St. Louis, MO, USA). The tissue was then homogenized using a tissue dissociator (GentleMACS Dissociator, Miltenyi Biotec, Bergisch Gladbach, Germany). Subsequently, the homogenate was centrifuged at 4 °C for 10 min at 13,000 rpm, and the supernatant was stored at −80 °C for further analysis.

Cytokine levels were assessed using the MILLIPLEX^®^ Mouse Th17 Magnetic Bead Panel kit (Merck^®^), according to the manufacturer’s instructions. The following molecules were evaluated: IL-2, TNFα, IL-1β, IL-6, IL-4, IL-10, and IL-17A, which are representative of different helper T cell cytokine profiles.

### 2.11. Statistical Analysis

The normality of the variables was assessed using the Shapiro–Wilk test prior to data analysis. Data were expressed as the mean ± standard error of the mean (SEM). To identify differences among experimental groups, two-way ANOVA was performed, followed by Dunnett’s or Tukey’s multiple comparison tests, as appropriate. *p*-values ≤ 0.05 were considered statistically significant [25]. Data analysis was conducted using GraphPad Prism 10.4 statistical package (GraphPad Software, San Diego, CA, USA).

## 3. Results

### 3.1. Body Weight and Intestinal Morphometry

Throughout the treatment schedule, the body weight of the animals was recorded weekly, as this parameter may correlate with CRC progression and serves as a critical indicator of health and well-being. The results demonstrate a reduction in body weight at the end of each DSS cycle in the positive control and both FMT groups, relative to the negative control. However, the animals exhibited weight recovery approximately 7 days following each cycle (Figure 2A).

Intestinal weight was also evaluated as a relevant parameter, as it may reflect the severity of injury in the gastrointestinal tract.

At week 12, an increase in intestinal weight was observed in the CRC-induced groups compared to the negative control (Figure 2B). However, statistical analysis revealed a significant increase in the FMT-Healthy group compared to the negative control.

In addition, colon length was assessed as an indicator of structural alterations associated with inflammation and tissue damage. At week 5, a statistically significant reduction in colon length was observed in the positive control, FMT-CRC, and FMT-Healthy groups compared to the negative control. At week 12, a significant decrease was still evident in the positive control relative to the negative control (Figure 2C).

The negative control group, which was not subjected to CRC induction, did not develop tumors. In contrast, tumor presence was confirmed in all other experimental groups, with no significant differences between those that received FMT and the positive control group. However, a trend toward increased tumor count was observed in the FMT groups compared to the positive control, suggesting a potential influence of FMT on tumor progression (Figure 2D).

Regarding morphometry, macroscopic lesion analysis revealed a higher proportion of lesions at week 5 in the CRC-induced groups, highlighting the effectiveness of the AOM/DSS model as a potent inducer of colorectal carcinogenesis. By week 12, more pronounced alterations were evident, with the FMT-CRC group exhibiting the greatest degree of lesions. These findings illustrate the progressive structural changes in the colon throughout the experiment (Figure 3).

### 3.2. Histopathology Evaluation

Histopathological analyses were performed on the mid and distal colon regions of the experimental groups at weeks 5 and 12. These analyses were essential to thoroughly assess structural and cellular tissue changes.

Histopathological analyses were conducted on the mid and distal regions of the colon from the experimental groups in weeks 5 and 12 of the experiment. These assessments were essential for evaluating structural and cellular changes in the tissues. A total of 31 histopathological parameters were examined, providing a comprehensive overview of the impact of human FMT on CRC development and progression. The evaluated parameters included: epithelial atrophy, glandular atrophy, goblet cell loss, epithelial hyperplasia, glandular hyperplasia, mitosis, apoptosis, aberrant crypts, epithelial dysplasia, glandular dysplasia, neoplastic proliferation, neutrophil infiltration, hyperplasia of gut-associated lymphoid tissue (GALT), erosion, ulceration, presence of pigment and minerals, necrosis, congestion, edema, hemorrhage, thrombi, fibrin, exocytosis, eosinophil, macrophage, lymphocyte, and plasmocyte infiltration, fibrosis, and presence of biological agents. The severity of lesions was quantified using an ordinal scale, allowing for comparative analysis across the different experimental groups throughout this study. Among the 31 parameters assessed, representative changes were observed only in neutrophils, macrophages, lymphocytes, fibrosis, neoplastic proliferation, mitosis, apoptosis, epithelial hyperplasia, dysplasia, and glandular damage.

At week 5, histopathological analysis of the mid-colon region (Figure 4A) revealed that the FMT-CRC Mice group exhibited a significant increase in mitotic activity, neoplastic proliferation, and vascular congestion compared to the control group. Additionally, greater mitotic activity, neoplastic proliferation, GALT hyperplasia, and lymphocyte infiltration were observed in comparison with the FMT-Healthy Mice group. By week 12, the FMT-CRC Mice displayed a significant increase in mitotic activity, apoptosis, neoplastic proliferation, GALT hyperplasia, congestion, and plasma cells infiltration compared to the positive control group. Furthermore, differences in mitosis, neoplastic proliferation, GALT hyperplasia, congestion, and plasma cells were noted between the FMT-CRC and FMT-Healthy Mice groups.

On the other hand, in the distal colon region at week 5 (Figure 4B), the FMT-CRC Mice exhibited higher mitotic activity and neoplastic proliferation compared to the FMT-Healthy Mice and the positive control group. Additionally, differences in necrosis were observed between the FMT-CRC and FMT-Healthy Mice groups. Higher levels of glandular dysplasia and congestion were also found in the FMT-Healthy Mice group compared to the positive control, whereas fibrosis was significantly more pronounced in the positive control group compared to both FMT groups.

At week 12, mice that received FMT, regardless of the microbiota origin, exhibited increased mitotic activity and neoplastic proliferation compared to the positive control. Additionally, greater GALT hyperplasia was observed in the FMT-CRC Mice compared to the FMT-Healthy Mice.

Regarding intestinal morphology assessed by Hematoxylin and Eosin staining (Figure 5), the negative control group displayed preserved epithelial architecture, with well-organized crypts and no visible inflammatory infiltrates. In contrast, the groups subjected to CRC induction with AOM/DSS (positive control, FMT-CRC group, and FMT-Healthy group) exhibited evident structural alterations, characterized by disruption of glandular architecture, increased mitotic activity, distorted crypts, areas of dysplasia, and atypical cellular proliferation. Additionally, some groups showed signs of GALT hyperplasia and inflammatory changes in the lamina propria. These histopathological features were more pronounced in week 12 compared to week 5.

### 3.3. Inflammatory Cytokine Determination

To understand the impact of the inflammatory response, measurement of inflammatory cytokines was performed as described in the methods. Supernatants from colonic tissue samples collected at weeks 5 and 12 were analyzed to assess the levels of key cytokines involved in the regulation of Th1, Th2, and Th17 immune responses, including IL-2, TNFα, IL-1β, IL-6, IL-4, IL-10, and IL-17A.

At week 5, a significant decrease in IL-2 levels were observed in the positive control and FMT-CRC groups compared to the negative control. Additionally, at the same time point, a significant increase in IL-10 levels was detected in the FMT-Healthy group compared to the positive control. By week 12, IL-2 levels were significantly lower in all CRC-induced groups compared to the negative control. Furthermore, a significant increase in IL-17A levels was found in the FMT-CRC group compared to both the negative and positive control groups (Figure 6).

No significant differences were observed between the positive and negative control groups in the cytokines, except for IL-2, where an increase was noted in the colon of the negative control mice. On the other hand, no statistical differences were found between the cytokine levels in the FMT-Healthy and FMT-CCR groups. However, for IL-10, a significant difference was observed at 5 week between the FMT-Healthy group and the other groups (Figure 6). Detailed statistical analyses for each cytokine at both time points are provided in the Appendix A.

## 4. Discussion

This study evaluated the histopathological and immunological effects of GM alteration on CRC progression using an AOM/DSS-induced murine model transplanted with human fecal microbiota. The AOM/DSS protocol is a well-established approach for inducing CRC and for assessing factors that modulate tumor development. As expected, mice exhibited body weight loss after each DSS cycle, followed by partial recovery during rest periods, reflecting the cyclic inflammatory response characteristic of this model [25]. DSS-induced inflammation is associated with disruption of the intestinal barrier, immune activation, and compensatory epithelial proliferation following tissue injury. In this context, the differences observed in intestinal morphometry among experimental groups suggest a modulatory effect of FMT on tissue remodeling processes [37,38].

Colon length and weight are widely used indicators of inflammation and pathological remodeling in AOM/DSS-based CRC models, as they indirectly reflect mucosal thickening, edema, and neoplastic transformation [39]. The significant reduction in colon length observed from week 5 in CRC-induced groups confirms the early structural impact of AOM/DSS treatment. This finding is consistent with previous reports describing colon shortening and increased tissue mass as hallmarks of chronic DSS-induced colitis and CRC development [40]. By week 12, this difference persisted only between positive control and negative control, while the FMT-treated groups no longer differed significantly from the negative control, suggesting a partial recovery of colon length potentially associated with FMT.

Regarding intestinal weight, all CRC-induced groups showed a tendency toward increased values compared to the negative control at week 12, consistent with progressive inflammatory-driven wall thickening. This trend was more pronounced in the FMT-treated groups, suggesting a potential interaction between the transplanted microbiota and the tumor microenvironment. Although no statistically significant differences were detected between FMT groups and the positive control, the FMT-Healthy group exhibited a slightly higher intestinal weight at week 12. This may reflect a more pronounced inflammatory response resulting from the interaction between microbiota from an apparently healthy donor and an already inflamed colonic environment, potentially exacerbating immune activation, altering tissue architecture, and promoting fibrosis. Such changes could contribute to increased tissue stiffness, colon shortening, and weight gain [40].

Similar morphological alterations have been reported in previous AOM/DSS studies. For instance, ref. [41] reported a 35% reduction in colon length following AOM/DSS administration compared to untreated animals. In contrast, studies using DSS-induced colitis models combined with FMT and anti-inflammatory interventions have demonstrated improved microbiota restoration and partial morphological recovery of the colon [42]. In the present study, BALB/c mice received FMT without concomitant anti-inflammatory treatment, which may explain why FMT did not fully counteract inflammatory damage, particularly at the early time point (week 5).

Histopathological analysis revealed significant differences in parameters related to cell proliferation, inflammation, and tissue remodeling, enabling evaluation of the impact of FMT on CRC progression. At week 5, mice receiving FMT from CRC patients showed increased mitotic activity and neoplastic proliferation in the mid-colon compared to both the positive control and FMT-Healthy groups, suggesting that CRC-derived microbiota may promote a more active tumor microenvironment at early disease stages. In addition, greater GALT hyperplasia and lymphocytic infiltration were observed in the FMT-CRC group relative to the FMT-Healthy group, indicating enhanced immune activation associated with the composition of the transplanted microbiota. More pronounced tissue congestion was also detected in this group, potentially reflecting vascular alterations linked to chronic inflammatory processes.

By week 12, the FMT-CRC group exhibited significantly higher mitotic activity, apoptosis, neoplastic proliferation, GALT hyperplasia, congestion, and plasma cell infiltration compared to the positive control, indicating progressive intensification of both proliferative and inflammatory processes as disease advanced. The persistence of differences between the FMT-CRC and FMT-Healthy groups further supports the hypothesis that microbiota derived from CRC patients promotes a colonic microenvironment more favorable to tumor progression.

Region-specific effects were also evident in the distal colon. At week 5, the FMT-CRC group displayed increased mitotic activity and neoplastic proliferation compared to both the FMT-Healthy and positive control groups. Differences in necrosis were observed between the FMT groups, while the FMT-Healthy group showed higher glandular dysplasia and congestion compared to positive control. In contrast, fibrosis was more pronounced in the positive control group, possibly reflecting distinct tissue remodeling responses in the absence of microbiota transplantation.

By week 12, mice receiving FMT—regardless of donor source—exhibited increased mitotic activity and neoplastic proliferation compared to the positive control, suggesting that FMT itself may influence proliferative pathways within the tumor microenvironment. Nevertheless, the FMT-CRC group showed greater GALT hyperplasia than the FMT-Healthy group, indicating sustained immune activation associated with chronic inflammation [43].

Comparable effects have been reported in chemically induced colitis models, such as those using 2,4,6-trinitrobenzenesulfonic acid (TNBS), where FMT from Crohn’s disease (CD) patients exacerbated intestinal and mesenteric inflammation, characterized by venous congestion, neutrophil margination, perivascular infiltration, and hematoma formation. Although FMT from healthy donors partially ameliorated these alterations, CD-derived FMT intensified tissue damage [44]. Despite methodological differences—such as the use of AOM/DSS in the present study and antibiotic-mediated microbiota depletion in the TNBS model, these findings underscore the context-dependent role of gut microbiota in driving inflammatory and neoplastic pathology.

Similarly, ref. [26] demonstrated that FMT from CRC patients increased histological inflammation and high-grade dysplasia compared to controls, although their analysis was limited to week 5 and included prior antibiotic treatment, which may have amplified the observed effects. Despite these differences, both studies consistently indicate that microbiota derived from CRC or inflammatory disease patients alter the colonic microenvironment in a manner that promotes inflammation, tissue damage, and tumor progression.

The analysis of pro- and anti-inflammatory cytokines provided insight into how the transplanted microbiota modulates immune regulation in chemically induced CRC. Chronic inflammation is a well-established driver of CRC initiation and progression. Pro-inflammatory cytokines, including IL-2, TNFα, IL-1β, and IL-6, contribute to shaping a tumor-promoting microenvironment by sustaining inflammation and facilitating cancer cell survival and dissemination [45]. Conversely, anti-inflammatory cytokines such as IL-4 and IL-10 can suppress immune surveillance while paradoxically supporting tumor progression through immunosuppressive mechanisms [46]. IL-17A further amplifies inflammation by inducing chemokine production, recruiting neutrophils, and activating T cells, thereby sustaining chronic inflammatory responses [47].

IL-10 displayed a distinctive temporal pattern in our study. At week 5, IL-10 levels were significantly increased in the FMT-Healthy group compared to the positive control, suggesting an early immunomodulatory effect of the transplanted microbiota. IL-10 plays a critical role in controlling excessive inflammation and protecting intestinal tissue from immune-mediated damage. By week 12, both FMT groups showed a trend toward increased IL-10 levels, which may reflect a host adaptive response to ongoing tumor progression. Although IL-10 is generally considered protective, in the context of cancer it may also contribute to the establishment of an immunosuppressive tumor microenvironment, highlighting its dual role during CRC progression.

These findings partially contrast with those reported by [48], who observed sustained increases in colonic IL-10 levels and IL-10–producing CD4^+^ T cells following FMT in a DSS-induced colitis model. Several methodological differences may account for this discrepancy. While their study focused on DSS-induced colitis, our model incorporated AOM-mediated genotoxic damage, which likely alters immune regulation within the tumor microenvironment. Additionally, they transplanted microbiota from normobiotic mice, whereas our study employed human-derived microbiota, which may exhibit reduced adaptability and functional integration within the murine intestinal niche.

The timing of cytokine assessment further complicates direct comparisons. In the colitis model, IL-10 increases were detected at early time points (days 11 and 15), whereas our study evaluated later stages (weeks 5 and 12), revealing a transient early increase that did not persist, underscoring the influence of experimental design on immunological outcomes.

Regarding IL-2, a consistent reduction was observed across CRC-induced groups compared to the negative control. At week 5, IL-2 levels were significantly decreased in the positive control and FMT-CRC groups, while by week 12, this reduction extended to all CRC-induced groups. IL-2 is essential for the expansion and activation of cytotoxic effector T cells; thus, its reduction suggests impaired antitumor immunity and the establishment of an immunosuppressive microenvironment. High IL-2 signaling promotes terminal effector differentiation and cytolytic function in CD8^+^ T cells [49]. Therefore, the observed IL-2 decline may facilitate tumor immune evasion by limiting effective cytotoxic responses.

These results partially align with findings by [50] who reported reduced IL-2 levels following FMT in a DSS-induced ulcerative colitis model. However, IL-2 levels in their UC model were elevated relative to negative controls, likely to reflect acute inflammation. In contrast, the AOM/DSS-induced CRC model represents a chronic tumor-driven process in which IL-2 suppression may be part of immune escape mechanisms [51,52]. Together, these observations suggest that FMT influences IL-2 signaling in a disease- and context-dependent manner.

A trend toward increased IL-17A levels was observed in both FMT groups at weeks 5 and 12, indicating that FMT itself may sustain inflammatory signaling regardless of donor source. At week 12, IL-17A levels were significantly higher in the FMT-CRC group compared to the positive control, reinforcing the hypothesis that CRC-derived microbiota contributes to a pro-inflammatory tumor microenvironment. In contrast, FMT from healthy donors reduced IL-17A levels in ulcerative colitis models, suggesting a protective effect in non-neoplastic inflammation. The absence of IL-17A reduction in the FMT-Healthy group in our study implies that FMT from an apparently healthy donor is insufficient to reverse Th17-driven inflammation in the context of established CRC. Factors such as tumor–microbiota interactions, microbial composition, and follow-up duration may explain this discrepancy.

Consistent with this interpretation, ref. [53] reported reduced pro-inflammatory cytokines following FMT in a chemotherapy-induced mucositis model. However, our findings diverge, as persistent trends toward increased IL-1β, IL-4, and IL-6 were observed in FMT-treated groups compared to the positive control. These differences likely reflect model-specific effects, as mucositis represents acute inflammation, whereas AOM/DSS-induced CRC involves chronic, multifactorial inflammation and tumorigenesis. Additional contributing factors include the origin of the transplanted microbiota (murine vs. human), the use of antibiotics to facilitate engraftment in mucositis models, and differences in cytokine assessment time points. Collectively, these factors may have attenuated potential anti-inflammatory effects of FMT or masked transient immunological changes in our model.

Beyond the immunological alterations observed, several well-established microbial signatures associated with CRC progression may help contextualize the immune and histopathological patterns identified in our model. For example, *Fusobacterium nucleatum*, frequently enriched in CRC patients, promotes tumorigenesis by activating NF-κB–mediated inflammatory pathways, enhancing IL-17A and IL-6 signaling, and suppressing cytotoxic T-cell activity through Fap2-dependent immune evasion mechanisms [54,55]. Similarly, other CRC-associated bacteria, such as enterotoxigenic *Bacteroides fragilis* and *Escherichia coli* strains harboring the *pks* genomic island, can induce DNA damage, disrupt epithelial integrity, and drive Th17-dependent pro-inflammatory responses.

Although the microbial composition of the fecal donors was not characterized in the present study, these well-described mechanistic pathways support the hypothesis that specific taxa enriched in CRC donors may have contributed to the increased mitotic activity, GALT hyperplasia, and pro-inflammatory cytokine profiles observed in the FMT-CRC group. Accordingly, future studies should incorporate 16S rRNA sequencing or metagenomic profiling to confirm microbiota engraftment, characterize donor-specific microbial signatures, and directly link microbial taxa to immunological and histopathological outcomes.

An additional limitation of this study relates to the use of human-to-mouse fecal microbiota transplantation. While this approach provides valuable translational insights, interspecies differences in intestinal physiology, immune system development, and host–microbe interactions may result in partial or selective microbial engraftment, potentially influencing immune modulation and limiting direct extrapolation of the findings to human CRC [56].

Finally, the relatively small sample size, with three animals per group at each time point, represents a further limitation. Although this number is commonly used in exploratory AOM/DSS-based CRC studies and was constrained by ethical considerations, it may reduce the statistical power to detect subtle effects, particularly in cytokine analyses. Nevertheless, the consistency of the histopathological alterations across experimental groups supports the biological relevance and robustness of the observed effects.

## 5. Conclusions

The findings of this study demonstrate that FMT influences CRC progression in a murine model induced with AOM/DSS. Body weight loss following DSS cycles, together with increased intestinal weight and reduced colon length, reflects the pathophysiological impact of the model and supports its utility in evaluating the effects of microbiota modulation. Although no significant differences in tumor count were observed between the FMT groups and the positive control, trends toward increased tumor burden in animals receiving human-derived microbiota suggest a potential microbial contribution to CRC progression.

At the histological level, FM particularly from CRC patients was associated with increased mitotic activity, neoplastic proliferation, and immune cell infiltration, as well as greater dysplasia and structural alterations in the mid and distal regions of the colon. These findings reinforce the hypothesis of a modulatory role of the gut microbiota in colorectal carcinogenesis. Notably, these alterations were more pronounced at week 12, suggesting a temporal progression of tissue damage.

Regarding the cytokine profile, notable modulations were observed in the levels of IL-1β, IL-4, IL-6, IL-10, IL-17A, and TNF-α, particularly in the FMT groups, indicating an influence on the local immune response that could contribute to shaping the tumor microenvironment. These findings suggest that human microbiota affects not only the architecture and function of colonic tissue but also the immunological mechanisms involved in tumorigenesis in this model. Taken together, these results support the relevance of the gut microbiota as a potentially determining factor in CRC progression and highlight the possibility of considering therapeutic strategies based on their modulation to prevent or limit the development of this pathology.

## Figures and Tables

**Figure 1 cimb-48-00015-f001:**
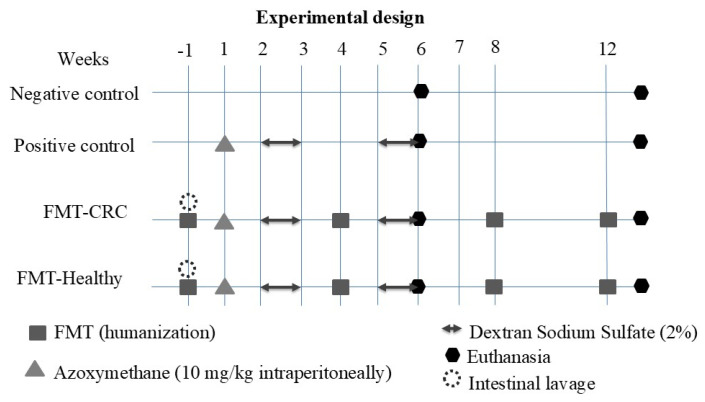
Experimental design.

**Figure 2 cimb-48-00015-f002:**
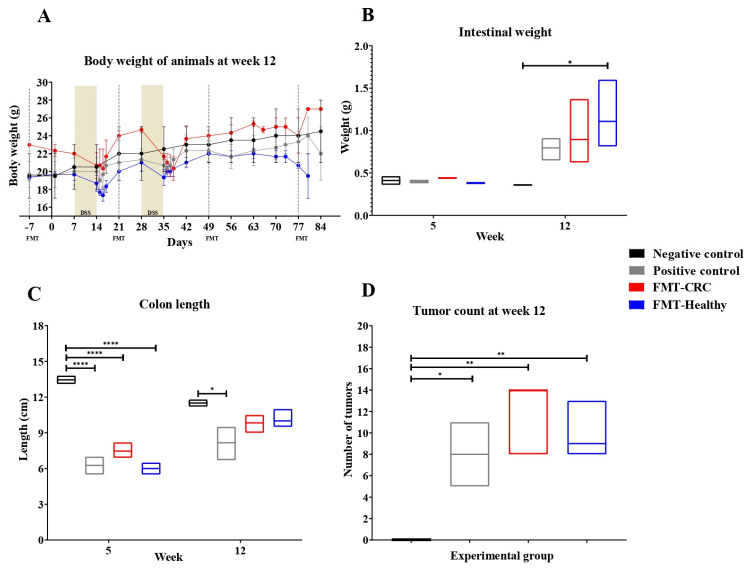
Effects of FMT on physiological and tumor parameters in a murine CRC model. (**A**) Body weight progression throughout the experiment. (**B**) Intestinal weight at weeks 5 and 12. (**C**) Colon length at weeks 5 and 12. (**D**) Tumor counts at week 12 in the different experimental groups. Data is presented as mean ± SEM. Statistical analysis was performed using two-way ANOVA. Statistical significance: * *p* < 0.05; ** *p* < 0.01; **** *p* < 0.0001 were considered significant.

**Figure 3 cimb-48-00015-f003:**
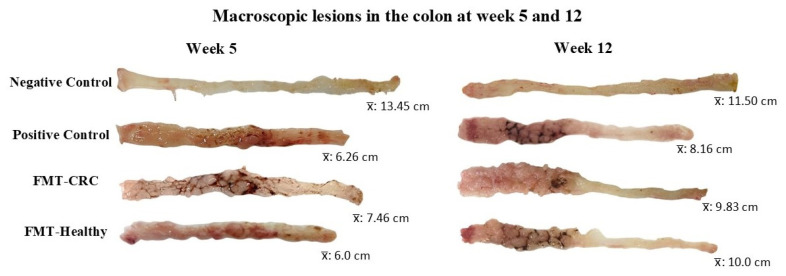
Representative photographs of macroscopic lesions in the colon at weeks 5 and 12, along with the average intestinal length of each experimental group.

**Figure 4 cimb-48-00015-f004:**
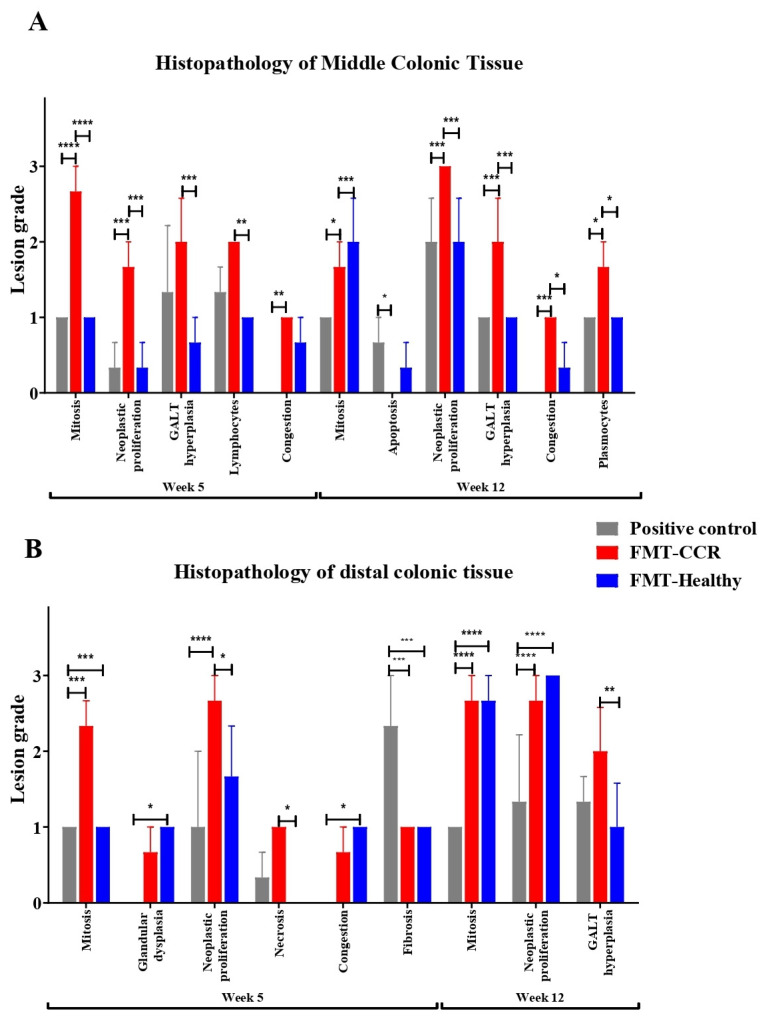
Histopathological lesions. (**A**) Middle region and (**B**) Distal region of the colon in the different experimental groups evaluated at weeks 5 and 12. Results are presented as mean ± SEM. Statistical analysis was performed using two-way ANOVA. Values of * *p* < 0.05; ** *p* < 0.01; *** *p* < 0.001; **** *p* < 0.0001 were considered statistically significant. Lesion grading: 0 = No lesion or absence of the condition, 1 = Mild to moderate lesion, 2 = Moderate lesion, 3 = Moderate to severe lesion. Only parameters showing statistically significant differences are presented. The negative control group was not included in the comparison of histopathological parameters, as the analysis focused on assessing statistical differences among groups induced with AOM/DSS.

**Figure 5 cimb-48-00015-f005:**
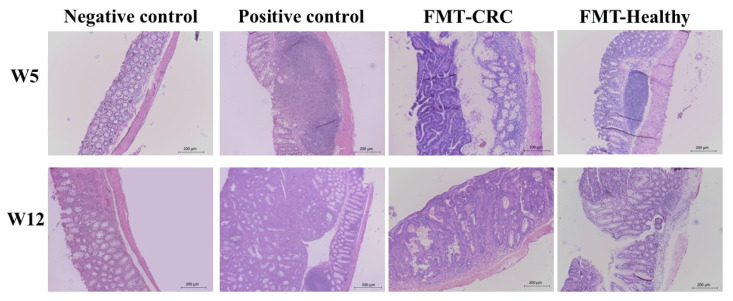
Representative histological images of colon sections stained with Hematoxylin-Eosin from the experimental groups at weeks 5 and 12. Additional histopathological parameters evaluated in this study are provided in the Appendix A.

**Figure 6 cimb-48-00015-f006:**
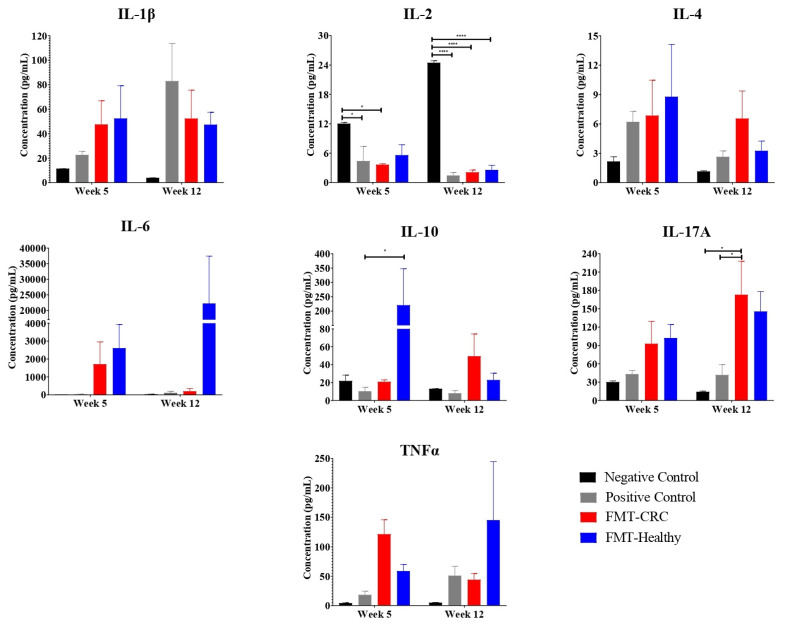
Evaluation of cytokine levels (IL-2, TNFα, IL-1β, IL-6, IL-4, IL-10, and IL-17A) in the supernatant of colonic tissue from the different experimental groups, assessed at weeks 5 and 12. Results are expressed as mean ± SEM. Statistical analysis was performed using two-way ANOVA. * *p* < 0.05; **** *p* < 0.0001 were considered statistically significant.

**Table 1 cimb-48-00015-t001:** Comparison of methodologies used for intestinal microbiota cleansing and fecal microbiota transplantation (FMT).

Reference	Mouse Strain	Age	Model	Microbiota Depletion			TMI		
				Method	Frequency	Duration	Dose	Frequency	Duration
[26]	C57BL/6	6 weeks	AOM	AB	Diary	2 weeks	200 μL (1 g/5 mL)	2 t/week	5 weeks
[27]	C57BL/6J	4 weeks	Apc min/+	AB	Diary	3 d	200 μL (1 g/5 mL)	1 t/d	1 week
								3 t/week	2 weeks
								1 t/week	3–8 weeks
[28]	C57BL/6J	5 weeks	AOM/DSS	AB	-	4 weeks	200 μL	3 t/week	Sacrifice (84 days)
[29]	C57BL/6	5 weeks	AOM/DSS	AB	-	4 weeks	-	Diary	12 weeks
[30]	C57BL/6 (GF)	8–10 weeks	AOM/DSS	N/A	N/A	N/A	200 μL	Days 1 and 5	N/A
[31]	Balb/c	6 weeks	AOM/DSS	-	-	-	200 μL	Days 14, 35 and 56	N/A
[32]	Balb/c (GF)	8–10 weeks	DSS	N/A	N/A	N/A	100 μL	Diary	7 days
[33]	C57BL/6J	4 weeks	N/A	AB	Diary	21 d	200 μL (0.1 g/L)	1 t/week	3 weeks
[34]	ICR	8 weeks	N/A	AB	-	2 weeks	200 μL (1 g/mL)	Diary	14 days
[35]	Balb/c	6 weeks	N/A	AB + PEG	Diary	3 d + 12 h PEG (10%)	200 μL (10 mg/mL)	1 dose	-
						3 weeks (Cycles 5/2 d)		5 doses	
[36]	C57BL/6J	3 weeks	N/A	AB + PEG	Diary	7 d + PEG (1 h later)	200 μL	Diary	3 days
	C57BL/6J (GF)	8 weeks	N/A	N/A	N/A	N/A			
[24]	C57BL/6J	8 weeks	N/A	PEG	20 min	1–6 doses	-	2 t/week	4 weeks
								1 t/week	4 weeks
								2 t/week	1 week
								1 t/week	1 week

PEG: polyethylene glycol; GF: germ-free; AB: antibiotic; t: times; d: day; (-): information not reported in the reference; N/A: not applicable.

## Data Availability

The original contributions presented in this study are included in this article/Appendix A. Further inquiries can be directed at the corresponding author.

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
