# Peer review of "Effect of Gut Microbiota Alteration on Colorectal Cancer Progression in an In Vivo Model: Histopathological and Immunological Evaluation"

_cimb, 2025, doi:10.3390/cimb48010015_

Round 1

Reviewer 1 Report

Comments and Suggestions for Authors This study investigates the influence of human gut microbiota (GM) on colorectal cancer (CRC) progression using a murine model. CRC was chemically induced via AOM/DSS in BALB/c mice, followed by fecal microbiota transplantation (FMT) from either healthy or CRC donors. Histopathological, morphometric, and immunological analyses, including cytokine profiling, were performed to evaluate tumor development and inflammatory responses. Results show that FMT from CRC donors accelerates histopathological features of CRC, increases mitotic activity, neoplastic proliferation, and immune cell infiltration compared to controls. Inclusion of both healthy and CRC FMT donors allows assessment of potential pro-tumorigenic versus protective roles of GM. Detailed descriptions of FMT procedures, intestinal lavage, and CRC induction support reproducibility, while combined morphometry, histopathology, and cytokine profiling provide a multidimensional evaluation of CRC progression.   I have identified these major issues:   1. The term “humanized” is inappropriate because the mice do not express human genes; FMT alone does not confer a humanized status. Remove this term.   2. Mechanistic links between GM, inflammation, T cell regulation, and tumorigenesis are not fully highlighted. Including microbiota composition (e.g., 16S rRNA or metagenomic profiling) would strengthen the mechanistic interpretation. At minimum, discuss known microbial differences (e.g., Fusobacterium nucleatum) relevant to CRC progression (PMID: 40227274, 41096975) in the main text, not just in limitations.   3. The table summarizing previous literature on intestinal microbiota cleansing and FMT should not appear in the Results section; it is more appropriately placed in Methods, as it provides procedural and contextual information rather than new findings.   4. Only three animals per group were euthanized at each time point. Discuss whether this sample size is sufficient for histopathological and cytokine analyses.   5. P-values are only described qualitatively in figures. Include exact p-values and effect sizes, at least for the most important data points.   6. Statements such as “FMT alone may promote cellular proliferation processes” are too vague and could imply causality without mechanistic evidence. Revise or clarify.   7. Terminology is inconsistent (e.g., “positive control” vs. “CRC-induced group”). Define group labels at first mention and maintain consistency.   8. The text briefly justifies the relevance of human-to-mouse FMT despite interspecies differences. Expand on this as a limitation in the Discussion section, considering potential impacts on microbiota engraftment and immune interactions.   9. The manuscript’s language is often redundant and convoluted. Simplify sentences for clarity and readability (e.g., “We used an AOM/DSS-induced CRC model, which we validated as an effective method to study colorectal cancer and the impact of FMT on tumor progression” instead of “For the execution of this research, a CRC model induced by AOM/DSS was used, which we validated as an effective method to induce colorectal cancer and as an experimental strategy to determine the effect of FMT on cancer progression.” In short, a general linguistic restyling would be very useful.

Author Response

  1. We thank the reviewer for this important clarification. We agree that the term “humanized” may be misleading in this context. Accordingly, we have removed the term throughout the manuscript and replaced it with more precise terminology referring to mice transplanted with human-derived fecal microbiota. This change has been implemented in the Introduction, Methods, Results, and Discussion sections.
  2. We appreciate this constructive suggestion. While microbiota composition was not assessed in this study, we have strengthened the Discussion by incorporating well-established mechanistic links between gut microbiota, inflammation, immune regulation, and CRC progression. Specifically, we now discuss CRC-associated microbial signatures such as Fusobacterium nucleatum, enterotoxigenic Bacteroides fragilis, and Escherichia coli harboring pks islands, and their roles in NF-κB activation, Th17-driven inflammation, immune evasion, and epithelial damage (PMID: 40227274; 41096975). This discussion has been added to the main Discussion section, rather than being limited to the study limitations. Additionally, we now explicitly state that future studies incorporating 16S rRNA sequencing or metagenomic profiling are necessary to confirm microbial engraftment and define causal mechanisms.
  3. We thank the reviewer for this suggestion. Following careful consideration of all reviewer comments, the table summarizing previous literature on intestinal microbiota depletion and fecal microbiota transplantation was removed from the Results section and its content was integrated into the Materials and Methods section. We believe this approach is appropriate because the information provides methodological context and procedural justification rather than experimental outcomes, and its inclusion in the Methods section improves clarity and reproducibility of the study.
  4. We have now discussed this limitation in the Discussion section, noting that the sample size was constrained by ethical considerations and may limit statistical power, particularly for cytokine analyses. However, consistent histopathological patterns were observed across animals, supporting the robustness of the main findings.
  5. We thank the reviewer for this suggestion. To improve statistical transparency and maintain the clarity of the main text, the exact p-values ​​and ANOVA statistics (F-values ​​and degrees of freedom) for all histopathological and immunological analyses have been included in the Supplementary Material.
  6. We have revised the Discussion to remove causal or overly general statements regarding the effects of FMT. The text now clarifies that FMT is associated with histopathological and immunological changes only within the context of an established AOM/DSS-induced inflammatory and tumor-prone microenvironment and does not imply that FMT alone acts as an independent driver of cellular proliferation.
  7. We thank the reviewer for this observation. We have clarified the terminology by explicitly defining “CRC-induced groups” at first mention as all AOM/DSS-treated groups, including the positive control, FMT-CRC, and FMT-Healthy groups, and we have ensured consistent use of these labels throughout the manuscript.
  8. We thank the reviewer for this important observation. We have expanded the Discussion to explicitly address the limitations associated with human-to-mouse FMT, including potential effects on microbiota engraftment and host immune interactions due to interspecies differences.
  9. We thank the reviewer for this suggestion. The Discussion section has been revised to improve clarity, conciseness, and readability by simplifying sentence structure, reducing redundancy, and adopting more direct language throughout the manuscript.

Reviewer 2 Report

Comments and Suggestions for Authors

The manuscript has good clinical value and provides more sufficient evidence for the influence of gut microbiota on the CRC process. The experimental design and results overall have strong persuasiveness. However, there are still some issues that need to be corrected to further improve the quality of the manuscript. The specific defects are as follows:

  1. If the author could add another group (Mice receiving human FMT from a patient diagnosed with CRC and without CRC induction using AOM/DSS) to compare with the negative control group, it would make the whole story more interesting.
  2. In section 3.2 "Histopathology evaluation", there are many dimensions of histopathology evaluation, and a supplementary data should be provided to show the details of intestinal pathological scoring for each mouse in each group, in order to reduce the subjectivity of scoring. There should also be relevant data for Negative control.
  3. Table 1 should be used as supplementary data instead of being included in the manuscript.
  4. There are still some simple errors in the manuscript that should be carefully checked. For example, line 165 clearly has an additional period. In addition, relevant ethical information need to be provided.

Author Response

  1. We appreciate the reviewer’s suggestion. While the inclusion of a group receiving human FMT from CRC patients without AOM/DSS induction would indeed be informative, this experiment falls outside the scope of the present study. The experimental design was completed as planned and the project has concluded, making the addition of new experimental groups unfeasible at this stage. Importantly, the primary objective of this work was not to assess the tumorigenic capacity of CRC-associated microbiota per se, but rather to evaluate how human-derived microbiota modulates disease progression within an established AOM/DSS-induced CRC model.
  1. We thank the reviewer for highlighting this point. In the negative control group, four animals were used in total (two at week 5 and two at week 12). Due to the limited thickness and fragility of colonic tissue, samples from individual animals were allocated exclusively to either histopathological analysis or cytokine measurement. Consequently, at each time point, one animal contributed to histopathological scoring and a different animal was used for cytokine analysis. This clarification has now been added to the Materials and Methods section, and individual data are transparently reported in the Supplementary Materials.
  2. We thank the reviewer for this suggestion. Following careful consideration of all reviewer comments, the table summarizing previous literature on intestinal microbiota depletion and fecal microbiota transplantation was removed from the Results section and its content was integrated into the Materials and Methods section. We believe this approach is appropriate because the information provides methodological context and procedural justification rather than experimental outcomes, and its inclusion in the Methods section improves clarity and reproducibility of the study
  3. We thank the reviewer for pointing out these issues. The manuscript has been carefully revised to correct typographical and formatting errors. In addition, relevant ethical information has now been clearly stated in the Methods section, including approval by the institutional ethics committee and compliance with applicable guidelines for animal experimentation.

Round 2

Reviewer 2 Report

Comments and Suggestions for Authors

After the author's revision, the manuscript has been greatly improved, and I have no further questions.